# Successful Emergency Management of a Dog with Ventilator-Dependent Acquired Myasthenia Gravis with Immunoadsorption

**DOI:** 10.3390/ani14010033

**Published:** 2023-12-21

**Authors:** Florian Sänger, Stefanie Dörfelt, Bettina Giani, Gesine Buhmann, Andrea Fischer, René Dörfelt

**Affiliations:** 1LMU Small Animal Clinic, Centre for Clinical Veterinary Medicine, Faculty of Veterinary Medicine, Ludwig-Maximilians-Universität München, 80539 Munich, Germany; f.saenger@medizinische-kleintierklinik.de (F.S.); b.giani@medizinische-kleintierklinik.de (B.G.); g.buhmann@medizinische-kleintierklinik.de (G.B.); andreafischer@lmu.de (A.F.); 2AniCura Veterinary Hospital Haar, 85540 Haar, Germany

**Keywords:** anti-acetylcholine receptor antibodies, megaesophagus, acute fulminating myasthenia gravis, generalized lower motor neuron, immunoadsorption, mechanical ventilation

## Abstract

**Simple Summary:**

Acquired myasthenia gravis is an immune-mediated neuromuscular disease caused by autoantibodies directed against the neuromuscular junction, which leads to skeletal muscle weakness. The acute fulminant form is defined as rapidly progressive generalized weakness and can lead to a life-threatening condition caused by the paralysis of respiratory muscles. Traditional management consists of acetylcholinesterase inhibitors and nutritional management. This case report shows a new therapeutic option for this acute life-threatening condition. The dog was mechanically ventilated for respiratory failure due to acquired myasthenia gravis and immunoadsorption, an extracorporeal blood purification technique, was performed. During immunoadsorption, blood is purified in an extracorporeal circuit using an adsorber that binds circulating autoantibodies. After immunoadsorption, the dog was successfully weaned from mechanical ventilation and further treated with medical management.

**Abstract:**

A one-year-old, female intact Samoyed, 12.5 kg, was presented with coughing for 2 weeks, progressive appendicular and axial muscle weakness, megaesophagus and labored breathing for 5 days. There was no improvement with standard treatment. Acquired myasthenia gravis was suspected and the dog was referred with increasing dyspnea. At presentation, the dog showed a severely reduced general condition, was non-ambulatory and showed abdominal and severely labored breathing. A marked hypercapnia (PvCO_2_ = 90.1 mmHg) was present in venous blood gas analysis. The serum anti-acetylcholine receptor antibody test was consistent with acquired myasthenia gravis (2.1 nmol/L). The dog was anesthetized with propofol and mechanically ventilated with a Hamilton C1 ventilator. Immunoadsorption was performed with the COM.TEC^®^ and ADAsorb^®^ platforms and a LIGASORB^®^ adsorber to eliminate anti-acetylcholine receptor antibodies. Local anticoagulation was performed with citrate. Treatment time for immunoadsorption was 1.5 h with a blood flow of 50 mL/min. A total plasma volume of 1.2 L was processed. Further medical treatment included intravenous fluid therapy, maropitant, esomeprazole, antibiotic therapy for aspiration pneumonia and neostigmine 0.04 mg/kg intramuscularly every 6 h for treatment of acquired myasthenia gravis. Mechanical ventilation was stopped after 12 h. A percutaneous gastric feeding tube was inserted under endoscopic control on day 2 for further medical treatment and nutrition. A second treatment with immunoadsorption was performed on day 3. Again, a total plasma volume of 1.2 L was processed. Immediately after this procedure, the dog regained muscle strength and was able to stand and to walk. After 6 days, the dog was discharged from the hospital. This is the first report of immunoadsorption for emergency management of a dog with acute-fulminant acquired myasthenia gravis. Immunoadsorption may be an additional option for emergency treatment in dogs with severe signs of acquired myasthenia gravis.

## 1. Introduction

Acquired myasthenia gravis (MG) is an autoimmune disease affecting neuromuscular transmission and it is one of the most common neuromuscular disorders in dogs [1,2,3]. In this immunologic disease, typically circulating immunoglobulin G autoantibodies, which are directed against a dwindling number of nicotinic acetylcholine receptors (AChR) at the postsynaptic membrane of the neuromuscular junction result in impaired neuromuscular transmission from block alteration or complement-mediated decay of acetylcholine receptors [4,5]. The consequence is skeletal muscle weakness, which can be focal, affecting only the facial, ocular, esophageal, pharyngeal, or laryngeal muscles. The generalized form affects also appendicular muscles with fatigability of muscle strength and exercise-induced weakness. Acute fulminant myasthenia gravis causes rapidly progressive generalized weakness, which can progress to respiratory failure [6]. In dogs, autoantibodies against muscle-specific tyrosine-kinase, and titin and ryanodine receptors are also described on rare occasions [7,8]. Megaesophagus with signs of regurgitation and aspiration pneumonia is a common complicating feature of acquired myasthenia gravis in dogs. The treatment goal of acquired myasthenia gravis in dogs is to manage the clinical signs and support survival until spontaneous clinical remission occurs. Aspiration pneumonia and respiratory failure are common causes of death in dogs with acquired MG, and regurgitation was negatively associated with clinical remission in a recent study [3,6,9,10,11]. The common treatment options for myasthenia gravis are the application of acetylcholinesterase inhibitors which increase the half-life of acetylcholine in the synaptic cleft and immunosuppressive therapy, which is primarily offered to dogs that fail to respond to standard treatment and are not suffering from aspiration pneumonia [3,10,12,13]. In human medicine, other treatment options targeting autoantibodies are high-dose human intravenous immunoglobulins (hIVIG) or therapeutic plasma exchange [14]. Human IVIG administration is also described in dogs with acquired MG [5,15]. The use of hIVIG in this disease in dogs is still debatable [5]. Therapeutic plasma exchange (TPE) is a treatment modality that can cause rapid improvement in patients with severe antibody-mediated immunologic diseases. The patient’s plasma, containing autoantibodies besides other substances, such as immune complexes and toxins, is replaced with replacement fluids, commonly donor plasma [16,17,18,19,20,21]. Four dogs with acquired MG which were managed with TPE are reported in the veterinary literature [18,22,23]. Two of these dogs showed clinical improvement after TPE and two other dogs died [18,23]. The latter had acquired MG secondary to neoplasia, i.e., multiple abdominal and cranial mediastinal masses, and hemangiosarcoma and thymoma [22,23]. Another treatment option for dogs with acquired MG described in the literature is MG vaccines. These vaccines contain peptides that are mimetics of antigen receptors on certain autoreactive T and B cells which produce the anti-AChR antibodies [24]. Using these vaccines, a greater rate of remission is possible in dogs with MG [24].

Immunoadsorption (IA) is an extracorporeal blood purification technique that is used to eliminate autoantibodies, primarily IgG, and circulating immunocomplexes from the blood [25]. The IA machine contains a centrifuge for separating erythrocytes and plasma. After separation, the plasma is transferred through an adsorber, which contains specific ligands that selectively bind antibodies and immunocomplexes [25]. In human medicine, IA is described as an alternative treatment in patients with acquired MG [26]. The indication for IA is a severe myasthenic crisis, or before surgery in patients with thymoma [26]. In veterinary medicine, IA was successfully used in the management of immune-mediated hemolytic anemia in a dog [27]. To the best knowledge of the authors, IA has yet not been reported in the veterinary literature for the treatment of acquired MG. This case report should highlight the first successful application of IA in a dog with acquired MG.

## 2. Case Presentation

A one-year-old, female intact Samoyed, weighing 12.5 kg, was presented due to coughing for two weeks, progressive weakness of the pelvic limbs which progressed towards non-ambulatory tetraparesis within one week, generalized megaesophagus (Figure 1) and labored breathing for five days. Aspiration pneumonia was suspected by the referring veterinarian and symptomatic treatment with amoxicillin/clavulanic acid and pyridostigmine bromide was started. As the dog deteriorated in the following days, it was referred with the suspicion of acquired MG and the presence of severe, life-threatening dyspnea. At presentation, the dog was recumbent and showed a severely reduced general condition. During physical examination, a heart rate of 70/min, hyperemic mucous membranes with a capillary refill time of <1 s, hyperthermia of 39.5 °C, respiratory rate of 20/min and labored abdominal breathing with signs of intercostal respiratory muscle weakness were detected. Neurologic examination showed a flaccid non-ambulatory tetraparesis with only minimal voluntary movements and absence of segmental spinal reflexes of all limbs (flexor withdrawal, patellar, and cranial tibial), consistent with a generalized lower motor neuron neuroanatomic localization. The initial laboratory examination showed a respiratory acidosis with severe hypercapnia (PvCO_2_ = 90.1 mmHg) and an increased C-reactive protein (CRP) of 89.0 mg/L (reference range < 10 mg/L). Metabolic causes for generalized weakness were ruled out by unremarkable hematology, serum biochemistry, electrolytes, T4 and cortisol measurement. For further investigation, electromyography and nerve conduction studies were performed. Electromyography showed no abnormal spontaneous activity in appendicular, paravertebral, and facial muscles. In motor nerve conduction studies, the amplitude of compound muscle action potentials measured after distal stimulation of the tibial nerve were small (2 mV, reference range 10.1–32.1 mV). Motor nerve conduction velocity was within normal limits (63 m/s). In repetitive nerve stimulation (frequency 3 Hz), the dog showed a decremental response of 64.9% comparing compound muscle action potentials of I and V amplitude. Serum anti-acetylcholine receptor (AChR) antibodies were tested with a new radioimmunoassay (ACHRAB^®^ RIA, Biocontrol, Ingelheim, Germany) and were increased (2.1 nmol/L; reference range ≤ 1.0 nmol/L), which confirmed the diagnosis of acquired MG [28]. Diagnostic imaging revealed no signs of neoplasia and history did not show any signs of an underlying process, so a primary immune-mediated disease was suspected.

Initial treatment consisted of endotracheal intubation and mechanical ventilation due to severe dyspnea and hypercapnia. Anesthesia was induced with propofol (Narcofol 10 mg/mL, cp-pharma, Burgdorf, Germany) and maintained with a propofol continuous rate infusion (0.1–0.3 mg/kg/min). Depth of anesthesia was adjusted to the patient’s requirement to induce the spontaneous initiation of the respiratory cycle by the patient. Mechanical ventilation was provided with the mechanical ventilator Hamilton C1 (Hamilton Medical AG, Bonaduz, Switzerland) using a pressure-supported sustained intermediated mandatory ventilation mode (PSIMV). Initial settings were an inspiratory pressure (Pinsp) of 5 mmHg, a positive end-expiratory pressure (PEEP) of 5 mmHg and a fraction of inspired oxygen (FiO_2_) of 50%. Settings (FiO_2_ and pressure support) were adapted during ventilation to maintain normoxemia, with oxygen saturation (SpO_2_) of 95–98%, and normocarbia (etCO_2_ 35–45 mmHg). FiO_2_ could be stepwise reduced to 30% during the ventilation period.

For further treatment and in an attempt to rapidly decrease the circulating anti-AChR antibody load, immunoadsorption was performed with a double-lumen central venous catheter with 12 Fr. and 20 cm (Arrow Germany GmbH, Erding, Germany), which was inserted in the right jugular vein. Continuous plasma separation with centrifugation was performed with the COM.TEC platform (Fresenius Kabi, Deutschland GmbH, Bad Homburg, Germany) using a commercial tubing system (P1R 9 400 411, Fresenius Kabi GmbH, Bad Homburg, Germany). After separation, the plasma was transferred to the connected immunoadsorption machine (ADAsorb, Medicap clinic GmbH, Ulrichstein, Germany). For immunoadsorption, a commercial tubing system (ADAsorb-LIGASORB Set, Medicap clinic GmbH, Ulrichstein, Germany) and a staphylococcus protein A column (LIGASORB, Fresenius Medical Care Deutschland GmbH, Bad Homburg, Germany) were used (Figure 2). Anticoagulation with citrate (ACD-A, Fresenius Kabi, Bad Homburg, Germany) at a blood/citrate ratio of 1/24 according to citrate anticoagulation protocols established for hemodialysis in dogs was used. Calcium gluconate 10% (Calciumgluconat 10%, B. Braun Melsungen GmbH, Melsungen, Germany) was infused via a peripheral venous catheter to avoid hypocalcemia. Ionized calcium measurements were performed every 30 min to achieve an ionized calcium concentration of <0.3 mmol/L in the extracorporeal circuit and >0.8 mmol/L in the patient. Extracorporeal blood volume was 405 mL. Tubing was filled with isotonic saline to flush the system. After that, prefilling with 200 mL of blood-group-matched stored whole blood was performed to reduce risks of hypovolemia and hemodilution due to the large extracorporeal blood volume. Due to limited resources and to avoid hemoconcentration after reinfusion, complete priming of the extracorporeal system with whole blood was not performed. A blood flow of 50 mL/min and plasma flow of 30 mL/min were selected to process 1.2 L of plasma during two IA cycles with 600 mL each (double plasma volume). Monitoring with electrocardiography, capnography, pulse oximetry and non-invasive blood pressure was performed during the procedure. Vital parameters were stable the whole time, except for a moderate hypothermia of 35.5–36.0 °C. Finally, 2 L of blood and 1.2 L of plasma were processed in two cycles of 600 mL each. Mechanical ventilation could be discontinued 12 h after IA. The dog was still non-ambulatory and required oxygen supplementation.

Further medical management consisted of intravenous fluid therapy, maropitant (1 mg/kg IV q 24 h; Prevomax, Dechra Pharmaceuticals, Northwich, UK), esomeprazole (1 mg/kg IV q 12 h; Nexium; Grünenthal, Aachen, Germany), amoxicillin/clavulanic acid (20 mg/kg IV q 8 h; Amoxiclav Hexal 500/100 mg, Hexal, Holzkirchen, Germany) and neostigmine (0.04 mg/kg IM q 6 h; Carinopharm, Eime, Germany). On the second day, a percutaneous endoscopic gastrostomy (PEG) tube was placed under general anesthesia for enteral nutrition and medication. On the third day, a second IA was performed. The same setting as during the first IA was used and 1.2 L of plasma was processed. After the second IA, the patient regained muscle strength and was able to stand and walk. Treatment for acquired MG was changed to pyridostigmine bromide (0.6 mg/kg PO q 8 h; Mestinon 10 mg, MEDA Pharma GmbH & Co. KG, Bad Homburg, Germany) after the second IA. No adverse effects were observed and the dose was increased to 0.8 mg/kg PO q 8 h on the following day. Serum AChR antibodies were analyzed after IA and on the following days. After the initial treatment, antibodies decreased by 75% (Figure 3). After six days, the patient was discharged from the hospital. At this time point, the clinical examination was unremarkable except for rare spontaneous coughing because of aspiration pneumonia, which was already improving. The dog was now able to walk for several minutes without signs of fatigue while in the neurologic examination, the dog still showed signs of generalized muscle weakness, i.e., decreased flexor withdrawal strength.

The dog was further treated by the referring veterinarian. Muscle strength and ability to walk were maintained but the megaesophagus did not improve and the dog developed several additional episodes of aspiration pneumonia while the PEG tube was still in place. This might be due to poor compliance of the pet owner with the feeding protocol, who continued to feed the dog orally. As AChR antibodies increased again, another cycle of IA was recommended but refused by the owners. Finally, the dog was euthanized due to the owner’s request, 3.5 months after the initial presentation during another episode of aspiration pneumonia.

## 3. Discussion

This is the first report of immunoadsorption (IA) in a dog with acute fulminant myasthenia gravis (MG) and severely labored breathing, requiring ventilatory support. In the present case, breathing pattern and effort improved markedly within a short time after the first IA so that the dog could be weaned from the ventilator after 12 h. Dogs with severe generalized lower motor neuron disease may require mechanical ventilation, or even die or are euthanized due to complications such as pneumonia, ventilation-associated lung injury, cardiac arrest, pneumothorax, or relapse of severe signs of weakness or aspiration pneumonia [15]. In one retrospective study of 14 dogs with generalized lower motor neuron disease, the median ventilation time was 109 h (range 5–261 h), which is longer than in the present case [15]. Acquired MG was diagnosed in 5/14 dogs and ventilation times specifically in these dogs were five hours, 33.4 h, 187 h, 192 h, and 261 h. Four of the five dogs died or were euthanized during hospitalization. The dog with a ventilation time of 192 h survived to discharge. In the present case, a marked clinical improvement was noticed after the second IA, as the dog regained muscle strength and was able to stand and walk again.

Different treatment options exist to deal with autoantibody formation in acquired MG and neuromuscular transmission disorders. Acquired MG is an autoimmune T-lymphocyte-dependent disease with the production of specific, typically immunoglobulin G, autoantibodies which are directed against nicotinic acetylcholine receptors (AChR) or other muscle proteins at the postsynaptic membrane of the neuromuscular junction [10,16,23,29]. The autoantibody production is caused by activation of CD4^+^ T (helper) lymphocytes and interaction with B lymphocytes [10]. In general, the cornerstone of therapy is the application of acetylcholinesterase inhibitors to decrease the breakdown of acetylcholine (ACh) and increase the half-life of ACh within the synaptic cleft [9,10,30,31]. Another approach is immunosuppression. Corticosteroids are used as first-line therapy for a short period in human medicine [32]. The precise mechanism of action in acquired MG is unknown. One mechanism includes the inhibition of T-cell activation by inhibiting the activation process in the cell nucleus and inhibiting antigen processing [32]. Corticosteroids may also improve the long-term prognosis in dogs with acquired MG cases but may also increase skeletal muscle weakness by causing dose-dependent exacerbation of weakness and steroid myopathy after long-term treatment [9,10,33]. In the case of aspiration pneumonia, the use of corticosteroids is contraindicated in human patients [34]. Also in dogs, the advantages and disadvantages of their use needs to be carefully weighed up in individual cases [3]. Other immunosuppressive drugs, which are considered in dogs with acquired MG, include azathioprine, cyclosporine, mycophenolate mofetil and leflunomide [10,12,35,36,37,38].

In human medicine, the management of patients with acute or severe MG is critically important because, otherwise, respiratory failure, paralysis and potential death can occur [39]. These patients are commonly treated with therapeutic plasma exchange (TPE) and high-dose human intravenous immunoglobulin (hIVIG) to stabilize the disease for a short time [39,40,41]. Recent human guidelines recommend the use of TPE in cases of severe acute MG, including myasthenic crisis, and before thymectomy [39]. TPE stabilizes acute patients as it removes the patient’s plasma, containing destructive autoantibodies, immune complexes, and toxins. Then, the patient’s plasma is replaced with substitution fluid, commonly donor plasma, potentially promoting transfusion reactions. [21,39,42,43]. Further recommendations to optimize TPE include early rather than delayed initiation of TPE during the crisis and delay of corticosteroids during the crisis until an improvement of clinical signs is observed after TPE [21,44].

In the veterinary literature, four canine cases with acquired MG exist, which were managed with TPE [18,22,23]. In 2/4 dogs, marked clinical improvement was noticed after TPE [23]. Both dogs were diagnosed with non-paraneoplastic generalized acquired MG. The treatment besides TPE included acetylcholinesterase inhibitors and immunosuppressive drugs (*n* = 1 mycophenolate mofetil; *n* = 1 corticosteroid) [22,23]. TPE was performed twice in both dogs 48 h apart. A subsequent antibody reduction of 50% and 70% was noticed. Of the two dogs that died, one had multiple abdominal and cranial mediastinal masses and died of multiple organ failure, secondary to aspiration pneumonia and sepsis. An antibody reduction of 82% was evident after two TPE sessions 48 h apart [23]. The other dog suffered from hemangiosarcoma and thymoma and underwent cardiopulmonary arrest during the first TPE. The dog was successfully resuscitated but required mechanical ventilation [18]. Even after the second TPE, the dog remained ventilated and developed oliguric acute kidney injury and was euthanized. In that dog, the AChR antibodies were reduced by 88%. Depending on local governmental restrictions, the major problem for TPE in dogs is the availability of canine plasma. Additionally, the use of plasma could be associated with transfusion reactions.

Another treatment option is human immunoglobulin application, which is also discussed in dogs with acquired MG [15]. The immune modulatory effect of hIVIG has not yet been fully elucidated but is thought to be caused by the following mechanisms: (I) Fc receptor disruption, (II) neutralization of pathologic autoantibodies, (III) complement inhibition, (IV) Fas-Fas ligand binding interference, and (V) cytokine synthesis downregulation [5,45,46,47,48,49,50,51,52]. TPE and hIVIG are often used interchangeably in human patients [39]. Human studies comparing TPE and hIVIG have not found significant differences in electrophysiological improvement, decrease in AChR antibody titer and quality of life [53,54,55]. However, there exists indirect evidence that the effect on the response time of TPE may be faster but less durable compared to hIVIG [54,56]. There is also evidence that patients treated with TPE need shorter ventilation times but longer hospitalization [39,56,57]. To date, the use of hIVIG in dogs with acquired MG is still debatable and the data are not sufficient to comment on any benefits [5,10].

The dog in the present case report showed rapidly progressive signs of skeletal muscle weakness, which deteriorated within two weeks to a flaccid lower motor neuron tetraparesis with only minimal voluntary movements, respiratory compromise and megaesophagus, classifying as acute fulminant presentation of MG regarding the classification system of veterinary medicine. Acute fulminant MG is present in <5% of dogs with generalized MG and represents an emergency case with acute, rapidly progressive, and very severe signs of generalized MG, in which respiratory failure and death can occur [10,33,58]. The acute fulminant form is still included in the current classification system in veterinary medicine. As the differentiation between generalized and fulminant presentation can be difficult as there are no objective criteria to differentiate these two categories, the term is no longer used in human medicine. In humans, severe generalized MG which starts with or progresses to severe skeletal muscle weakness to a point that requires intubation and mechanical ventilation is termed “myasthenic crisis” [59,60,61]. As the dog of the present case report deteriorated to an emergency patient, resembling a myasthenic crisis in human medicine, needing intubation and mechanical ventilation, another treatment modality, immunoadsorption (IA), with a potential for rapid improvement was advised.

IA is a procedure which allows the selective removal of humoral factors like immunoglobulins or complement factors [26]. IA uses a specific ligand, the adsorption column, which can bind ligates like pathogenic antibodies or immunocomplexes [25,26]. IA is well described in human medicine for the treatment of MG [26,62,63]. Usually, IA is used for the stabilization of patients in myasthenic crises to manage the life-threatening period. After that, medical management with prednisolone or other immunosuppressants and acetylcholinesterase inhibitors is still necessary [26]. Most patients require serial sessions of IA to treat myasthenic crises and 1.5–2.0 L of plasma is processed in humans during a single session of IA [26,62,63]. In human patients, a reduction of serum AChR antibodies to 23% of initial levels is achieved [62]. Human studies comparing TPE to IA in acute acquired MG found no significant differences between these two treatments, but the combination of TPE with IA was associated with a significantly shorter hospitalization time than TPE alone [64,65]. Another study observed that IA was not effective in all cases of acquired MG. Different subtypes of IgG are present in patients with acquired MG and the adsorption columns for IA are not able to bind all of these subtypes of IgG [26]. In the veterinary literature, IA has yet not been described for the treatment of neurological disorders. A case report of successful treatment of a dog with immune-mediated hemolytic anemia is the only reference applying IA in dogs [27].

In the present case, a reduction of AChR antibodies to 23.8% of the initial concentration immediately after IA in association with rapid improvement of weakness was achieved. Antibody reduction occurred again after the second IA. This reduction ratio is similar to that in human studies and indicates that IA could have a similar efficacy in dogs as in humans. Clinical improvement in patients with myasthenic crises usually occurs within 48 h after the first IA session in human patients [62,63]. In the present case, clinical improvement was evident after 12 h as the patient was weaned from mechanical ventilation. After two IA sessions (approximately 48 h after initial presentation), the patient was able to stand and walk again. However, improvement could also have been caused by the simultaneous intramuscular administration of neostigmine. Nevertheless, the decrease in AChR antibodies suggests a pronounced effect of IA on serum anti-AChR antibody load in this dog and that IA likely contributed to the pronounced clinical improvement and successful emergency management of this dog. IA might be a preferable technique to plasma exchange for emergency management of the myasthenic crisis in the future, as replacement plasma is not required [62]. Anti-AChR antibodies increased again after IA. Thirteen days after presentation, anti-AChR antibody concentration nearly reached its initial concentration and further increased in the following weeks. In human patients, anti-AChR antibody concentration increases after immunoadsorption, if no further treatment is provided, because the underlying disease process is not solved and plasma cells still produce antibodies [62]. Therefore, immunosuppressive treatment should be considered in dogs with acquired MG after IA. Nevertheless, the long-term prognosis of MG in dogs still relies on the successful management of the megaesophagus and avoidance of regurgitation and aspiration pneumonia as seen in this case. Pet owners need to be advised that the outcome could still be fatal despite extensive feeding management, e.g., feeding in upright positions, Bailey chair and PEG tube or esophageal suction, but in uncomplicated cases, spontaneous remission occurs in more than 50% of dogs in a median time of six months.

## 4. Conclusions

IA is a potential emergency treatment for dogs with acute-fulminant acquired MG as a supportive treatment option. AChR antibody concentration can be significantly reduced and, therefore, ventilation time could be shortened. Additional medical management with acetylcholinesterase inhibitors and immunosuppressants is still required. Further studies are needed to evaluate the effect of IA in dogs with acquired MG.

## Figures and Tables

**Figure 1 animals-14-00033-f001:**
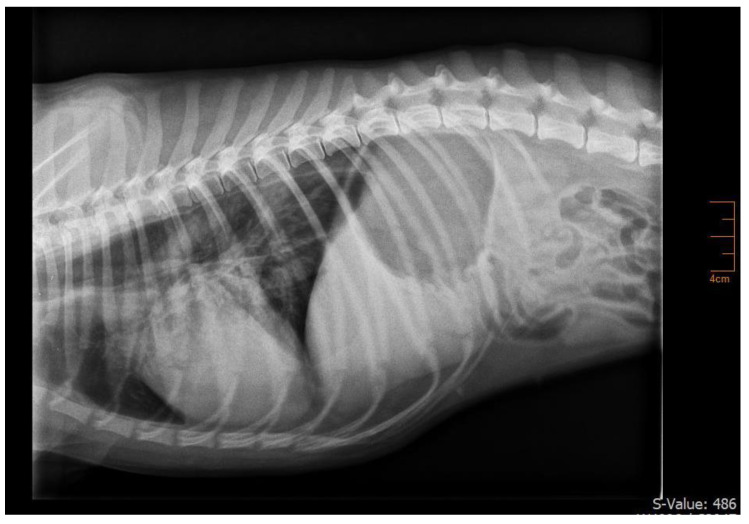
Thoracic radiograph of the dog with acute fulminating acquired myasthenia gravis during initial diagnostic workup. A generalized megaesophagus can be seen.

**Figure 2 animals-14-00033-f002:**
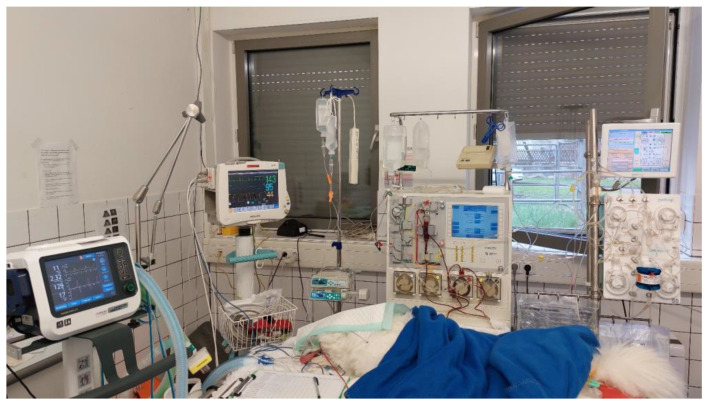
Treatment setup with immunoadsorption and mechanical ventilation for the dog with acquired myasthenia gravis.

**Figure 3 animals-14-00033-f003:**
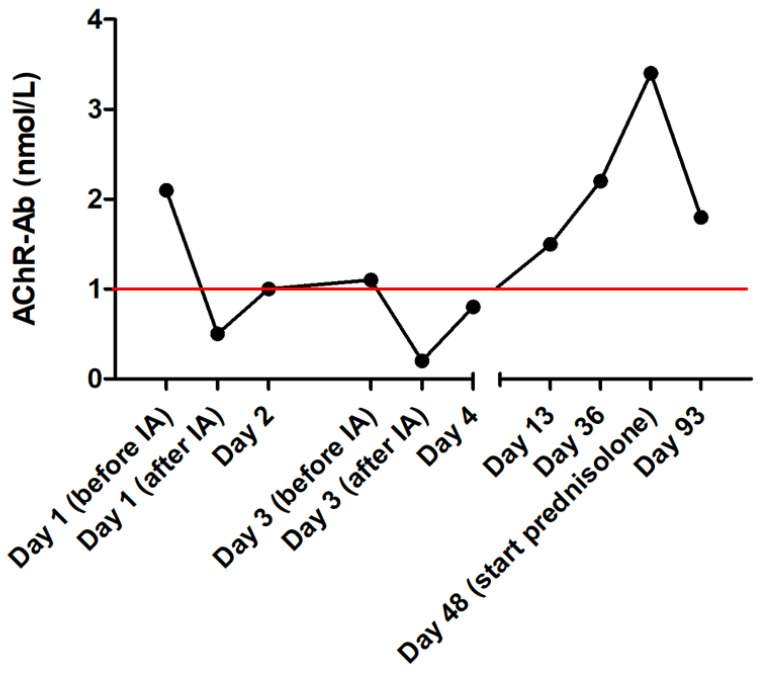
Results of anti-acetylcholine receptor antibodies in a dog with acute fulminating acquired myasthenia gravis treated with immunoadsorption (IA), analyzed with radioimmunoassay, before and after IA. The red line marks the upper reference range.

## Data Availability

The data presented in this study are available on request from the corresponding author.

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
