# Peer review of "Successful Emergency Management of a Dog with Ventilator-Dependent Acquired Myasthenia Gravis with Immunoadsorption"

_animals, 2023, doi:10.3390/ani14010033_

Round 1

Reviewer 1 Report

Comments and Suggestions for Authors

Dear authors, I have carefully reviewed the current case presentation. An interesting case, where utilization of an extracorporeal therapy complementary approach resulted in the effective management of a myasthenic crisis. However, I have identified some issues that need to be adressed.

1. Lines 123-124: Propofol was used for the induction of anaesthesia. How was the anaesthesia maintained throughout the 12-hour mechanical ventilation period? Which anaesthetic agents wer utilized?

2. Lines 124-126: Which were the mechanical ventilation parameters (you should mention, at least, the parameters you utilized for the induction of the mechanical ventilation considering your pressure ventilation mode)?

3. Lines 129-145: This part is totally copied by another case report. Rephrase. Did you use exactly the same setting for your 12.5kg dog, with that of Richter et al.(2021) where the dog was 28kg (concerning the circuit volume)?

Richter, P, Fischer, H, Dörfelt, R. Immunoadsorption in a dog with severe immune mediated hemolytic anemia. J Clin Apher. 2021; 36(4): 668–672. https://doi.org/10.1002/jca.21913

4. Was there any adverse effects noticed during the extracorporeal therapy? What about Blood pressure? SPO2?

5. Taking into account that the extracorporeal circuit volume was 405ml, it seems that almost half of the blood of the 12kg-dog was outside of the dog during the procedure. Was there a need for administration of any blood products? What was the priming solution of your circuit? Just saline? 

6. What about measurements of ionized calcium during the procedure? We do know that you face hypocalcemia to some extent, because of citrate anticoagulation. Did you measure ionized calcium in the dog and in the circuit in order to adjust the citrate administration, or in order to adjust the calcium administration during the procedure?

Author Response

Dear reviewer,

Thank you for your input. We tried to address your comments and adapted the manuscript accordingly. Please find our comments below.

Reviewer 1

Dear authors, I have carefully reviewed the current case presentation. An interesting case, where utilization of an extracorporeal therapy complementary approach resulted in the effective management of a myasthenic crisis. However, I have identified some issues that need to be adressed.

  1. Lines 123-124: Propofol was used for the induction of anaesthesia. How was the anaesthesia maintained throughout the 12-hour mechanical ventilation period? Which anaesthetic agents wer utilized?

→ Anesthesia was maintained with a propofol CRI for the whole time. No other anesthetic medications were used.

  1. Lines 124-126: Which were the mechanical ventilation parameters (you should mention, at least, the parameters you utilized for the induction of the mechanical ventilation considering your pressure ventilation mode)?

→ Added to the text

  1. Lines 129-145: This part is totally copied by another case report. Rephrase. Did you use exactly the same setting for your 12.5kg dog, with that of Richter et al.(2021) where the dog was 28kg (concerning the circuit volume)?

Richter, P, Fischer, H, Dörfelt, R. Immunoadsorption in a dog with severe immune mediated hemolytic anemia. J Clin Apher. 2021; 36(4): 668–672. https://doi.org/10.1002/jca.21913

→ Circuit volume and blood flow rate are standardized for immunoadsorption and are independent of the patient’s body weight. Therefore, the settings were the same as for the 28 kg patient.

  1. Was there any adverse effects noticed during the extracorporeal therapy? What about Blood pressure? SPO2?

→ Blood pressure and SpO2 have been in the normal range. Only hypothermia was observed. Added to the text.

  1. Taking into account that the extracorporeal circuit volume was 405ml, it seems that almost half of the blood of the 12kg-dog was outside of the dog during the procedure. Was there a need for administration of any blood products? What was the priming solution of your circuit? Just saline?

→ After flushing the circuit with saline, it was prefilled with whole blood. Added to the text.

  1. What about measurements of ionized calcium during the procedure? We do know that you face hypocalcemia to some extent, because of citrate anticoagulation. Did you measure ionized calcium in the dog and in the circuit in order to adjust the citrate administration, or in order to adjust the calcium administration during the procedure?

→ Ionized calcium measurements in the system and patient were performed. Added to the text.

Reviewer 2 Report

Comments and Suggestions for Authors

Manuscript is interesting and raises a novel and potential emergency treatment for dogs with acute fulminating acquired MG as a supportive treatment option.   

Yet, it suffers from some minor problems; most of them can easily be solved by revising the manuscript. 

Simple summary, abstract and keywords:

The keywords section should be revised: words such as immunoadsorption and ventilator must be included, while word such: immunosuppressive treatment and plasma exchange, should be omitted since no immunosuppressive treatment was given and a specific plasma exchange (immunoadsorption), was used.  

Introduction:

In general, this section is well written. Yet, I believe it can be improved as followed:

1.      I strongly suggest elaborating the last paragraph dealing with the immunoadsorption (suggested reference: Willi Paul, Chandra P. Sharma. 27 Aug 2015, Plasma Perfusion: Immunosorbent Applications from: Encyclopedia of Surface and Colloid Science, Third Edition CRC Press).

2.      Please include a paragraph regarding anti canine myasthenia gravis vaccines as a possible immunotherapy (suggested ref: Galin FS, Chrisman CL, Cook JR Jr, Xu L, Jackson PL, Noerager BD, Weathington NM, Blalock JE. Possible therapeutic vaccines for canine myasthenia gravis: implications for the human disease and associated fatigue. Brain Behav Immun. 2007 Mar;21(3):323-31. doi: 10.1016/j.bbi.2006.10.001. Epub 2006 Nov 20. PMID: 17113748; PMCID: PMC1857319).

3.      This section should end by informing the reader about the objective of this case report (please revise the text, in order the objective/s to be clear).   

Case presentation:

·        Please add the reference interval for the AChR antibodies concentration to line 117; I also suggest, adding a cutoff line (which represents normal upper reference concentration) to figure 3.

·        Since myasthenia gravis can be secondary immune mediated (due to primary etiologies such as: thymoma, vaccines, drugs etc.) and for the completion of the case report presentation, please add refer the relevant history/diagnosis to determine whether the disease in this case is primary or secondary immune mediated.

Discussion

Please refer to the increase in AChR antibodies concentration, post 13 days from 2nd AI to similar level observed at day 1 prior to 1st AI, as the effectiveness of the AI in reducing these antibodies. Moreover, to the fact that as long as anti AChR producing plasma cell exists, antibody concentration will be building up and repeated AI will be needed.

Author Response

Dear reviewer,

Thank you for your input. We tried to address your comments and adapted the manuscript accordingly. Please find our comments below.

Reviewer 2

Manuscript is interesting and raises a novel and potential emergency treatment for dogs with acute fulminating acquired MG as a supportive treatment option.  

Yet, it suffers from some minor problems; most of them can easily be solved by revising the manuscript.

Simple summary, abstract and keywords:

The keywords section should be revised: words such as immunoadsorption and ventilator must be included, while word such: immunosuppressive treatment and plasma exchange, should be omitted since no immunosuppressive treatment was given and a specific plasma exchange (immunoadsorption), was used. 

→ keywords changed

Introduction:

In general, this section is well written. Yet, I believe it can be improved as followed:

  1. I strongly suggest elaborating the last paragraph dealing with the immunoadsorption (suggested reference: Willi Paul, Chandra P. Sharma. 27 Aug 2015, Plasma Perfusion: Immunosorbent Applications from: Encyclopedia of Surface and Colloid Science, Third Edition CRC Press).

→ Text elaborated. Unfortunately, we were not able to get the suggested reference in the short time.

  1. Please include a paragraph regarding anti canine myasthenia gravis vaccines as a possible immunotherapy (suggested ref: Galin FS, Chrisman CL, Cook JR Jr, Xu L, Jackson PL, Noerager BD, Weathington NM, Blalock JE. Possible therapeutic vaccines for canine myasthenia gravis: implications for the human disease and associated fatigue. Brain Behav Immun. 2007 Mar;21(3):323-31. doi: 10.1016/j.bbi.2006.10.001. Epub 2006 Nov 20. PMID: 17113748; PMCID: PMC1857319).

→ Added to the text

  1. This section should end by informing the reader about the objective of this case report (please revise the text, in order the objective/s to be clear).

→ Added to the text

Case presentation:

  • Please add the reference interval for the AChR antibodies concentration to line 117; I also suggest, adding a cutoff line (which represents normal upper reference concentration) to figure 3.

→ Added to text and figure

  • Since myasthenia gravis can be secondary immune mediated (due to primary etiologies such as: thymoma, vaccines, drugs etc.) and for the completion of the case report presentation, please add refer the relevant history/diagnosis to determine whether the disease in this case is primary or secondary immune mediated.

→ Added to the text

Discussion:

Please refer to the increase in AChR antibodies concentration, post 13 days from 2nd AI to similar level observed at day 1 prior to 1st AI, as the effectiveness of the AI in reducing these antibodies. Moreover, to the fact that as long as anti AChR producing plasma cell exists, antibody concentration will be building up and repeated AI will be needed.

→ Added to the text

Round 2

Reviewer 1 Report

Comments and Suggestions for Authors

Dear authors.

The responses to my comments are deficient, or roughly added to the manuscript.

Furthermore, I am not quite sure if the methods described, or these modified after the 1rst round, were indeed taken into consideration when dealing with the case. For example, a blood priming according to such a long circuit utilized, was not a detail that could be inadvertently omitted. In addition to that, I want you to know that I am well aware of the parameters and the procedure of immunoadsorption (according to your response on my comment about plagiarism detection).

My 1rst round comments, are not fully answered, and those that were taken into consideration were very roughly and abruptly done.

1. Details were asked for the anesthesia maintenance during mechanical ventilation (no dosages or range of constant rate infusion was mentioned)

2. You added to the text a vague report for the mechanical ventilation settings. You mention "Settings were adapted during ventilation due to the patient’s individual needs''... What individual needs? I think I get your point but if someone wants to repeat part of it, he won't be able to do it. You should mention for example "in order to maintain normocapnia" or " PECO2 between 35-45 mmHg''..

Comments on the Quality of English Language

Minor editing

Author Response

Dear reviewer,

We are sorry that your comments were not included sufficiently and that the paper was not improved, as you mentioned. We tried to include additional information to your comments.

Kind regards

The responses to my comments are deficient, or roughly added to the manuscript.

Furthermore, I am not quite sure if the methods described, or these modified after the 1rst round, were indeed taken into consideration when dealing with the case. For example, a blood priming according to such a long circuit utilized, was not a detail that could be inadvertently omitted. In addition to that, I want you to know that I am well aware of the parameters and the procedure of immunoadsorption (according to your response on my comment about plagiarism detection).

You refer to the paper:

Richter, P, Fischer, H, Dörfelt, R. Immunoadsorption in a dog with severe immune mediated hemolytic anemia. J Clin Apher. 2021; 36(4): 668–672. https://doi.org/10.1002/jca.21913

→ This paper is also from our group and we used the same machine and the same settings as in this paper. We are sorry that not mentioning the blood priming in the first version. It was not recorded in the electronic patient records, so it was missed during the case description.

The text was adapted as follows.

Continuous plasma separation with centrifugation was performed with the COM.TEC platform (Fresenius Kabi, Deutschland GmbH, Bad Homburg, Germany) using a commercial tubing system (P1R 9 400 411, Fresenius Kabi GmbH, Bad Homburg, Germany). After separation, the plasma was transferred to the connected immunoadsorption machine (ADAsorb, Medicap clinic GmbH, Ulrichstein, Germany). For immunoadsorption, a commercial tubing system (ADAsorb-LIGASORB Set, Medicap clinic GmbH, Ulrichstein, Germany) and a staphylococcus protein A column (LIGASORB, Fresenius Medical Care Deutschland GmbH, Bad Homburg, Germany) were used (Figure 2). For regional anticoagulation, sodium citrate (ACD-A, Fresenius Kabi, Bad Homburg, Germany) at a blood/citrate ratio of 1/24 according to citrate anticoagulation protocols established for hemodialysis in dogs was used. Calcium gluconate 10 % (Calciumgluconat 10 %, B. Braun Melsungen GmbH, Melsungen, Germany) was infused via a peripheral venous catheter to avoid hypocalcemia according to the established regional citrate anticoagulation protocoll. Ionized calcium measurements were performed every 30 minutes to achieve an ionized calcium concentration of < 0.3 mmol/l in the extracorporeal circuit and > 0.8 mmol/l in the patient. Extracorporeal blood volume was 405 ml. Tubing was filled with isotonic saline to flush the system. After that, prefilling with 200 ml of blood group matched stored whole blood was performed to reduce risks of hypovolemia and hemodilution due to the large extracorporeal blood volume. Due to limited resources and to avoid hemoconcentration after reinfusion, complete priming of the extracorporeal system with whole blood was not performed. Blood flow of 50 ml/min and plasma flow of 30 ml/kg/min were selected to process 1.2 l of plasma during two IA cycles with 600 ml each (double plasma volume).

  1. Details were asked for the anesthesia maintenance during mechanical ventilation (no dosages or range of constant rate infusion was mentioned)

The information on Propofol CRI was added:

Anesthesia was induced with propofol (Narcofol 10 mg/ml, cp-pharma, Burgdorf, Germany) to enable intubation and maintained with a propofol continuous rate infusion (0.1 – 0.3 mg/kg/min). The depth of anesthesia was adjusted to the patient’s requirement to provide spontaneous initiation of the respiratory cycle by the patient.

  1. You added to the text a vague report for the mechanical ventilation settings. You mention "Settings were adapted during ventilation due to the patient’s individual needs''... What individual needs? I think I get your point but if someone wants to repeat part of it, he won't be able to do it. You should mention for example "in order to maintain normocapnia" or " PECO2 between 35-45 mmHg''.

Additional information on ventilator setting were added.

Settings (FiO2 and pressure support) were adapted during ventilation to maintain normoxemia, with oxygen saturation (SpO2) of 95 – 98 %, and normocarbia (etCO2 35 – 45 mmHg). FiO2 could be stepwise reduced to 30 % during the ventilation period.

Round 3

Reviewer 1 Report

Comments and Suggestions for Authors

Thank you for your feedback.

Please incorporate the details requested in the second Review, in the manuscript (propofol CRI and mechanical ventilation details). I couldn't find them in final the re-submitted version.

As I have already mentioned in my 1st review, this is a very interesting study.

Good luck!

Comments on the Quality of English Language

Minor editing required.

Author Response

dear Reviewer,

Thank you for the comment. We assume to have uploaded the wrong version of the manuscript. Please find here the current version. Changes are marked in yellow. 

Kind regards